# Thermodynamic Functions of a Metal Exposed to High Energy Densities in Compressed and Expanded States

**Nikolay B. Volkov and Alexander I. Lipchak ***

Institute of Electrophysics the Ural Branch of the Russian Academy of Sciences, 620016 Yekaterinburg, Russia
* Correspondence: lipchak@iep.uran.ru

**Abstract:** The development of a wide-range phenomenological model of metal with a small number of adjustable parameters for studying the behavior of metals in expanded and compressed states under the exposition of high energy density fluxes is the objective of the paper present. Both the reference data, methods of the quantum-statistical model of the atom, the density functional theory, and the requirement to the expanded and compressed states description of metal should be consistent on their boundary were used in the model. The expressions for thermodynamic functions and the critical parameters of expanded iron were obtained within the framework of the soft sphere model. The Grüneisen parameters calculated for the expanded and compressed states of the metal appear to be in good agreement with each other was shown. A calculation technique of the ion component average charge of the metal in expanded and compressed states is proposed. The experimentally defined volume range of $V/V_0 = 3$–$4$ in which the character of iron conductivity changes from metallic to non-metallic includes the obtained in frameworks of our approach value of the critical volume: $V/V_0 = 3.802$ was established. The behavior of the average charge of the ion component is discussed. The contribution of the thermal electrons to the thermodynamic functions is evaluated.

**Keywords:** thermodynamic functions; high energy densities; metals; equations of state; liquid-vapor metal-insulator phase transitions; behavior of electrons near the critical region





## 1. Introduction

The action of high-intensity energy fluxes on a matter is widespread in nature and is used in practical applications of high energy density physics [1,2]. Under such influence, the state of matter changes dramatically in a broad range of temperatures and densities. The last one changes from an ideal gas to a highly dense substance. A complete range of problems arising in the theoretical and experimental study of matter behavior at temperatures and pressures intrinsic to high energy density physics and its applications is shown [2]. The point is that the particular interest, in this case, is the effect of intense laser radiation on a metal. The interest is associated with the presence of a free boundary in the irradiated metal, which expands to vacuum or cold gas at atmospheric or high pressure. The solid-liquid and liquid-vapor phase transitions (PT) accompany this expansion [1,2]. In addition, a rapid change in the electron concentration near the critical point of PT was observed [3]. It is a consequence of a sharp decrease in the overlap of the valence-shell electrons due to the metal-dielectric PT [4]. This situation is common for a high-pressure gas discharger controlled by laser radiation [5,6] when the last one forms initial plasma on an electrode, e.g., an anode. As a result, it expands to the cathode in an external electrostatic field.

The objective of the paper is a building of phenomenological model for a porous-free metal. We used iron as an example to determine its thermodynamic functions depending on temperature $T$ and specific volume $V$.

The contribution of thermally excited ions to these functions is considered within the Debye solid-state model [7,8], e.g., in shock wave physics. It is proposed to separate expanded and compressed matter states using this approach. While following this approach,

one has to consider the contribution of the thermal excitation of electrons to the thermodynamic functions of an expanding matter. For a compressed porous-free metal case, the Debye frequency $\omega_D$ and temperature $T_D = \hbar \times w_D/k_B$ depend only on its density $\rho$ where $k_B$ is the Boltzmann constant and $\hbar$ is the reduced Planck constant. As it will be shown below, the Grüneisen parameter $\gamma = \mathrm{d} \ln(\omega_D)/\mathrm{d} \ln(\rho)$ tends to its limiting value of 2/3 for high compressions. Results of the experimental study of melting and structural PT of simple metals during their quasi-static compression in Bridgman diamond anvils [9–14] are an additional argument in favor of Debye frequency independent of temperature. In [15,16], the insufficiency of a single-phase description of melting based on the Lindemann criterion, which is used to identify melting in shock wave experiments [17], was shown.

Similarly to [18], the model of soft spheres in a range of expanded matter is used to describe the thermal contribution of ions to thermodynamic functions. W. Hoover [19] and D. Young [20] proposed this model as a generalization of the classical Van der Waals equation. The advantage of this model is the absence of description difficulties of the free space. In addition, in the model of soft spheres, the volume coincides with the volume of a quasi-neutral atomic cell attributed to one atom, i.e., a central ion with an electron density neutralizing its charge, as it can be also seen in a compressed substance. This approach allows one to calculate the contribution of electrons to thermodynamic functions in a similar way both for the expanded state and compressed ones of a metal.

We calculated the contribution of the thermal excitation of electrons to thermodynamical functions using a density functional theory method (DFT) for $T = 0$ involving the LmtART-7 software developed by Prof. S. Yu. Savrasov [21,22]. The results of these calculations determined the contribution of electrons. Moreover, we used the results of calculations involving the quantum-statistical model of the atom [23,24] in the range of high values of relative densities 10–20. This allowed us to propose corrections to the chemical potential of the quantum-statistical model of the atom at $T = 0$, providing the correct description of the behavior of the electronic component in the region of low and normal metal densities. As a result, one could calculate the thermodynamic functions of the electronic and the ion components and the Grüneisen parameters for expanded and compressed matter states. Also, we paid attention to the behavior of thermodynamic functions near the critical point of the ion component and the boundary of the metal-dielectric PT.

## 2. Metal Models and Thermodynamic Functions

Commonly (see [2,7,15,18] and refs herein), the free energy for determining the thermodynamic functions of a continuous medium is written in the following form:

$$F(V,T) = E_c(V) + F_n(V,T) + F_{Te}(V,T) \tag{1}$$

The first term is the cold compression function, which accounts the interaction of the nuclei and all electrons in an atomic cell of volume $V_a$ at $T = 0$. The second one expresses the contribution to the thermodynamic functions of the thermal excitation of ions. The last term depicts the contribution of the thermal excitation of the electrons of the atomic cell. Due to the neutrality of the atom cell, the quantity of excited electrons is equal to the average charge of the ion. This value depends on the cell volume and temperature: $Z_i = \sum_{z=0}^{Z} Z n_z / n$, here $n = V_a^{-1} = \sum_{z=0}^{Z} n_z$, $V_a$ is the volume of the atom cell, and $Z$—nucleus charge [8]. Using known expressions of statistical physics [9] for pressure and internal energy, one can obtain from (1) the following expressions:

$$P(V,T) = -\frac{\partial F}{\partial V} = P_{cx}(V) + P_{Ti}(V,T) + P_{Te}(V,T); \tag{2}$$

$$E(V,T) = -T^2 \left( \frac{\partial}{\partial T} \frac{F}{T} \right)_V = E_{cx}(V) + E_{Ti}(V,T) + E_{Te}(V,T). \tag{3}$$

### 2.1. Cold Pressure and Energy

As a basis for obtaining cold compression functions, one can use the expression for the binding energy in the form of the Born-Mayer potential proposed in [25] (also see refs herein).

$$E_\pi(\delta) = \Lambda \overline{E}_\pi(\delta) = \Lambda \frac{1}{\alpha - 1} \left( \exp\left(\eta\sqrt{\alpha}\left(1 - \delta^{-1/3}\right)\right) - \alpha \exp\left(\frac{\eta\left(1 - \delta^{-1/3}\right)}{\sqrt{\alpha}}\right) \right), \quad (4)$$

where $\Lambda$ is the binding energy for $\delta = 1$, $\delta = \rho/\rho_0$, $\rho_0 = \rho(T = 0)$ (constants $\alpha$, $\eta$ see (7) below). In the range of applicability of quantum-statistical models (QSM) of an atom, expression (4) does not agree with the well-known fact that the energy and pressure of the degenerated electron gas determine the cold compression functions. Taking into account this fact and keeping the behavior of the cold compression functions to be almost unchanged, we suggest using the following relation instead of (4):

$$E_{cx}(\delta) = \Lambda \overline{E}_\pi(\delta) \frac{1 + a\delta^{1/3} + b\delta^{2/3}}{1 + a + b}, \quad (5)$$

here coefficients $a$, $b$, which one can find from the requirement of coincidence with the cold compression function calculated in the framework of the Thomas-Fermi model with quantum and exchange corrections (QSM) [26]. According to (5), the following expressions define the cold pressure:

$$P_{cx}(\delta) = \Lambda \rho_0 \left( \overline{P}_\pi(\delta) \frac{1 + a\delta^{1/3} + b\delta^{2/3}}{1 + a + b} + \overline{E}_\pi(\delta) \frac{a\delta^{4/3} + 2b\delta^{5/3}}{3(1 + a + b)} \right),$$
$$\overline{P}_\pi(\delta) = \frac{\sqrt{\alpha}\eta\delta^{2/3}}{3(\alpha - 1)} \left( \exp\left(\eta\sqrt{\alpha}\left(1 - \delta^{-1/3}\right)\right) - \exp\left(\frac{\eta\left(1 - \delta^{-1/3}\right)}{\sqrt{\alpha}}\right) \right). \quad (6)$$

The Grüneisen parameter relates with constants $\alpha$ and $\eta$, included in (4)–(6), by the following expression [24–26]:

$$\gamma_{\pi 0} = \frac{(\alpha + 1)}{6\sqrt{\alpha}} \eta. \quad (7)$$

One can use expression (7) to improve the accuracy of parameters $\alpha$ and $\eta$ using known, e.g., the experimental value of the Grüneisen parameter.

As one can see from (6), the relation for cold pressure $P_c(\delta = 1) = 0$, i.e., for $T = 0$, gives $a = -2b$. Using the tables from work [26], one can calculate the parameters $b = 6.021 \times 10^{-2}$ and, accordingly, $a = -1.204 \times 10^{-1}$ for the iron we investigated experimentally in [5]. For their definition, we used the values of the parameters we refined: $\alpha = 2.953$; $\eta = 5.156$. These values correspond to $\gamma_{\pi o} = 1.977$.

Figure 1a demonstrates the adequacy of the description of the quantum-statistical curve of cold compression using expressions (4)–(6). Figure 1b shows the matching of the energy in a compressed state at $T = 0$ to that one determined by the thermodynamic functions of the degenerate electron gas.

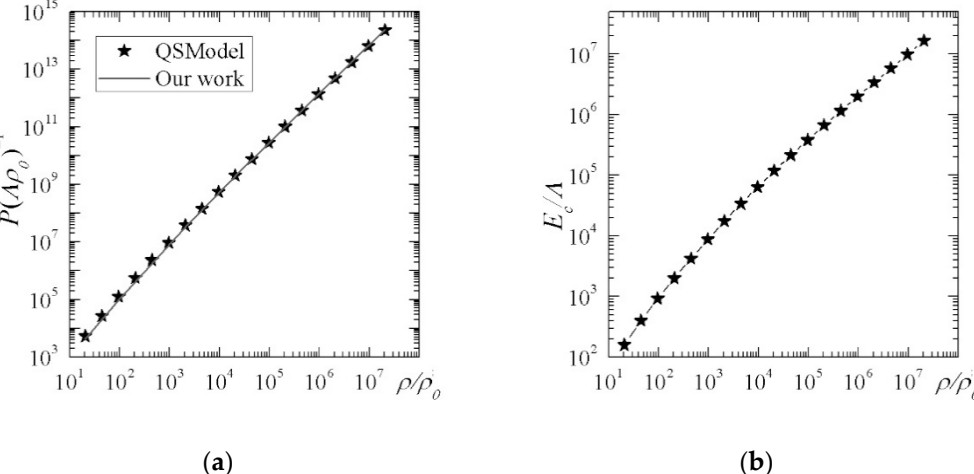

**Figure 1.** Calculations of cold compression: (**a**) comparison of the cold compression curve for the QSM model [25] and the proposed model, (**b**) matching of the energy of cold compression to the energy of a highly degenerate electron gas.

### 2.2. Cold Energy and Pressure in the Soft Sphere Model

The expanded state of matter differs from the compressed one in that it consists of singly charged ions, neutral atoms, and electrons at rather low densities and temperatures but still sufficient for ionization.

Solid-liquid and liquid-vapor PT accompany the transition of the metal from $\delta = 1$ densities to small values of $\delta$. The disappearance of the overlap of electron shells at temperatures close to zero causes metal-dielectric PT [4]. The last one takes place in the electron liquid. Assuming the solid-liquid and liquid-gas PT occur in an ionic liquid, i.e., the structural part of the free energy or the sum of the first two terms of (1). Thus, it seems to be incorrect to use expression (4) since it overestimates the contribution of the band structure of the metal in the case of its expanded state [25].

Let us use the well-described, e.g., in [18] method of generalized the Lennard-Jones type potentials formation to obtain the equation of state for cold extension (compression) in an expanded matter. Besides, it requires the continuity condition of the description of substance in expanded and compressed states for $\delta = 1$. As a result, we obtain the following relations:

$$E_{ex}(\delta) = \frac{\Lambda}{m - n}\left(n\delta^{\frac{m}{3}} - m\delta^{\frac{n}{3}}\right) \tag{8}$$

$$P_{ex}(\delta) = \frac{mn\Lambda\rho_0}{3(m - n)}\left(\delta^{(1+\frac{m}{3})} - \delta^{(1+\frac{n}{3})}\right); \tag{9}$$

$$c_{ex}^2 = \frac{dP_{ex}}{d\rho} = \frac{dP_{ex}}{\rho_0 d\delta} = \frac{mn\Lambda}{3(m - n)}\left(\left(1 + \frac{m}{3}\right)\delta^{\frac{m}{3}} - \left(1 + \frac{n}{3}\right)\delta^{\frac{n}{3}}\right). \tag{10}$$

Equations (8) and (9) satisfy conditions for the continuity of cold energy and pressure at $\delta = 1$. But Equation (10) needs to be verified. Differentiating the pair of Equation (6), we obtain $c_{cx}^2/\Lambda = 2.944$ for $\delta = 1$. Respectively, from (10), we get $c_{cx}^2/\Lambda = mn/9$. If one is limited using only integer values of $m$ and $n$, then the following values $m = 9$, $n = 3$ will be obtained. Equations (8) and (9) do not give a satisfactory description of the iron under compression so one should use Equations (5) and (6) to study expanded states. Figure 2a demonstrates the behavior of zero isotherms $P_c(\delta)$, $E_c(\delta)$ for $\delta < 1$, and Figure 2b $P_c(\delta)$, $E_c(\delta)$ for $\delta > 1$.

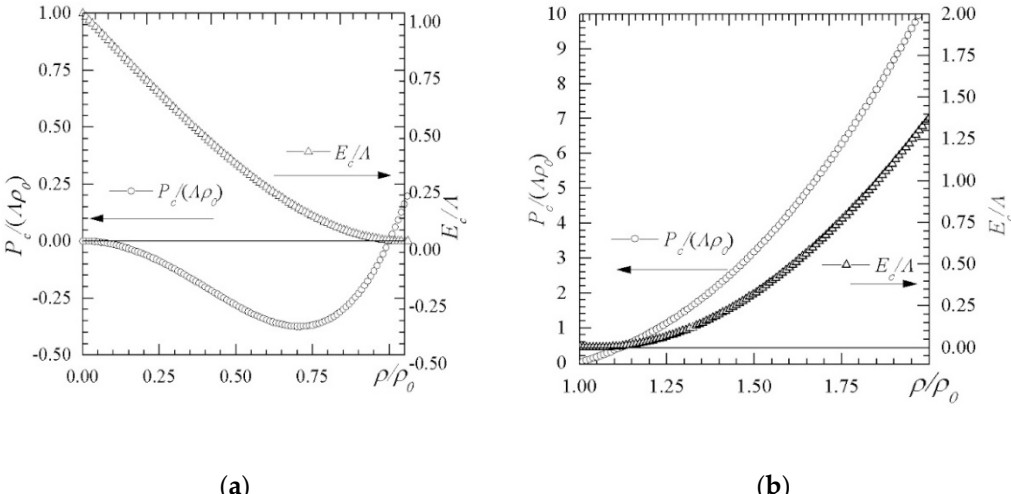

(**a**)                                             (**b**)

**Figure 2.** Zero isotherms: (**a**) expanded iron, (**b**) compressed iron.

### 2.3. Thermodynamic Functions of the Ion Component of a Metal

First, we consider the description of the thermodynamic functions of compressed metal. The study of structural and melting PT in the compressed state of metal requires a detailed consideration of the dynamics of the phonon and electronic spectra. The last one depends on the conditions of its loading and heating. We concentrate here on the expanded state of the metal. One can describe the compressed state using the Debye model [9]. For the expression of the compressed metal free energy, one can assume the lattice zero oscillations energy is included in that one of cold compression. Then the specific free energy of an excited simple lattice is determined by the following expression [9]:

$$F_{Tc}(\delta, \overline{T}) = \Lambda \overline{F}_{Tc}(\delta, \overline{T}) = \Lambda \overline{T}\left(3\ln\left(1 - \exp\left(-\frac{\overline{\omega}_D}{\overline{T}}\right)\right) - D\left(\frac{\overline{\omega}_D}{\overline{T}}\right)\right), \qquad (11)$$

here $\overline{\omega}_D = \hbar\omega_D / \Lambda_a$ is the dimensionless Debye frequency $\overline{T} = k_B T / (\hbar\omega_D)$, $\Lambda_a = \Lambda M_A$, $M_a = Au$ is the mass of the atom, $A$ is the atomic weight of the element, $u = 1.660 \times 10^{-27}$, kg is the atomic mass constant [27], and:

$$D(x) = \frac{3}{x^3}\int_0^x \frac{z^3 dz}{\exp(z) - 1} \quad \left(x = \frac{\overline{\omega}_D}{\overline{T}}\right) \qquad (12)$$

is Debye function.

Applying (11) known thermodynamic expressions, one can obtain the following:

$$E_{Tic}(\delta, \overline{T}) = \Lambda \overline{E}_{Tic}(\delta, \overline{T}), \quad \overline{E}_{Tic}(\delta, \overline{T}) = 3\overline{T}D\left(\frac{\overline{\omega}_D}{\overline{T}}\right) \qquad (13)$$

The relation (13) gives a contribution of thermal excitation of the ion component to the internal energy of the compressed metal.

$$P_{Tic}(\delta, \overline{T}) = \Lambda \rho_0 \overline{P}_{Tic}(\delta, \overline{T}), \quad \overline{P}_{Tic}(\delta, \overline{T}) = \Gamma(\delta)\delta\overline{E}_{Tic}(\delta, \overline{T}) \qquad (14)$$

The relation (14) gives a thermal pressure of the ion component. Here in (14), $\Gamma$ is the Grüneisen parameter of the ion component of the compressed metal.

One can see (11)–(14) that it is necessary to find an explicit expression for $\overline{\omega}_D(\delta)$ to calculate the thermodynamic functions and the Grüneisen parameter of the ion component for the compressed metal. In the Debye model of a solid, the mean sound speed determines the limiting frequency [9]:

$$\omega_m = \overline{c}_s \left( \frac{6\pi^2}{V_a} \right)^{1/3} = \overline{c}_s q, \; q = \left( \frac{6\pi^2}{V_a} \right)^{1/3}, \; V_a = VAu \tag{15}$$

By definition, the following relation determines the mean speed of sound [9]:

$$\frac{3}{\overline{c}_s{}^3} = \frac{2}{c_t{}^3} + \frac{1}{c_l{}^3}, \tag{16}$$

here the indexes "$t$" and "$l$" denote transverse and longitudinal sound wave speeds, respectively. Here we use the relations of Debye frequency for evaluation of the average sound speed proposed first by S. B. Kormer and V. D. Urlin [28,29] to interpret the results of experiments of shock-wave compression of porous metals and metal melting in a shock wave:

$$c_{cx} = \Lambda^{1/2} \left( \frac{d\overline{P}_{cx}(\delta)}{d\delta} - \frac{2}{3} n_1 \frac{\overline{P}_{cx}(\delta)}{\delta} \right)^{\frac{1}{2}}. \tag{17}$$

The second term in (17) makes it possible to consider the "softening" of phonon modes with pressure increase. Hence one can obtain an expression for the Debye frequency using (17) and (16):

$$\overline{\omega}_D(\delta) = \overline{\omega}_{mD} \delta^{\frac{1}{3}} \left( \left( \frac{d\overline{P}_{cx}}{d\delta} - \frac{2}{3} n_1 \frac{\overline{P}_{cx}}{\delta} \right) \Big/ \frac{d\overline{P}_{cx}}{d\delta} \Big|_{\delta=1} \right)^{\frac{1}{2}}, \; \overline{\omega}_D = \frac{\omega_D}{\omega_{D0}},$$
$$\overline{\omega}_{mD} = \frac{\omega_{TD}}{\omega_{D0}} = \frac{k_B \theta}{\hbar \sqrt{\Lambda} q_{s0}} = 1.319. \tag{18}$$

Taking the logarithmic derivative of (18), we obtain an expression for the Grüneisen parameter:

$$\Gamma(\delta) = \frac{1}{3} + \frac{1}{2} \frac{\delta \frac{d^2\overline{P}_{cx}}{d\delta^2} - \frac{2}{3} n_1 \left( \frac{d\overline{P}_{cx}}{d\delta} - \frac{\overline{P}_{cx}}{\delta} \right)}{\frac{d\overline{P}_{cx}}{d\delta} - \frac{2}{3} n_1 \frac{\overline{P}_{cx}}{\delta}} \tag{19}$$

It should be noted that relation (19) does not contain $\overline{\omega}_{mD}$. The shock compression of porous copper, aluminum, nickel, and lead was studied [28]. The first two have $n_1 = 0$, nickel has $n_1 = 1$, and the last one has $n_1 = -1$. Also, in his survey, L.V. Altshuler stated that the majority of metals has $n_1 = 1$ [30]. The thermodynamic similarity between metals of the same groups of the Periodic Table allows one to expect that for low-melting metals, $n_1 = 1$, and $n_1$ is equal to either 0 or 1 for other metals. By analogy with nickel for iron, we assume the parameter $n_1 = 1$.

The two ways of calculations according to (19) and (6) with the parameters of cold pressure are shown in Section 2.1 above. A simple calculation according to (6) gives the Grüneisen parameter $\Gamma(1) = 1.981$. It contains only $\alpha$ and $\eta$ parameters of the potential given by expression (4) and yields a minimal difference: $\Delta\Gamma = (\Gamma(1) - w\gamma_{\pi 0})/\gamma_{\pi 0} = 0.0024 << 1$.

The calculation technique used for the thermal contribution of ions to the thermodynamic functions of the metal in the expanded state is demonstrated below. One can use the soft sphere model based on classical works [19,20]. According to [18], the exponents $m$ and $n$ in (8)–(9) are related to parameters of Lennard–Jones type pair potential of atoms (ions):

$$U(r) = \varepsilon_{rep}\left(\frac{\sigma}{r}\right)^{\frac{m}{3}} - \varepsilon_{att}\left(\frac{\sigma}{r}\right)^{\frac{n}{3}} \tag{20}$$

here terms $\varepsilon_{rep}$ and $\varepsilon_{att}$ determine the repulsive and attractive parts of the interaction energy, respectively, according to the following relations:

$$\varepsilon_{rep} = \Lambda_a \frac{2}{C_m}\frac{n}{m-n}, \varepsilon_{att} = \Lambda_a \frac{2}{C_n}\frac{m}{m-n}. \tag{21}$$

here $\Lambda_a$ is the binding energy per ion (atom), $m = 9$, $n = 3$, and $C_m$, $C_n$ are lattice sums [31].

One can obtain the value of the dimensionless constant $\bar{\varepsilon}_{rep}^{(bcc)} = \varepsilon^{(bcc)}{}_{rep}/\Lambda_a = 0.1005$, $\bar{\varepsilon}_{rep}^{(fcc)} = \varepsilon^{(fcc)}{}_{rep}/\Lambda_a = 0.07952$ for *bcc* and *fcc* lattices of iron approximating these lattice sums. As for liquid metal, it is common to use the *fcc* lattice in the soft sphere model [19,20]. Also, one can express the contribution to the internal energy of expanded metal ions in frameworks of the soft sphere model as follows [18–20]:

$$E_{soft}(\rho,T) = \Lambda\bar{E}_{Tsoft}(\delta,\bar{T}), \ \bar{E}_{soft}(\delta,\bar{T}) = \bar{T}\left(\frac{3}{2} + \frac{1}{6}(m+4)\delta^{\frac{m}{9}}\left(\frac{\bar{\varepsilon}_{rep}}{\bar{T}}\right)^{\frac{1}{3}}Q\right) \tag{22}$$

$$P_{soft}(\delta,\bar{T}) = \Lambda\rho_0\bar{P}_{soft}(\delta,\bar{T}), \ \bar{P}_{soft}(\delta,\bar{T}) = \Gamma_{soft}(\delta,\bar{T})\delta\bar{E}_{soft}(\delta,\bar{T}) \tag{23}$$

The expression (23) is the thermal pressure of ions in the soft spheres model, and (24) is the Grüneisen parameter of the ion component of the expanded metal for this model.

$$\Gamma_{soft}(\delta,\bar{T}) = \left(1 + \frac{1}{18}Qm(m+4)\delta^{\frac{m}{9}}\left(\frac{\bar{\varepsilon}_{rep}}{\bar{T}}\right)^{\frac{1}{3}}\right)/\left(\frac{3}{2} + \frac{1}{6}Q(m+4)\delta^{\frac{m}{9}}\left(\frac{\bar{\varepsilon}_{rep}}{\bar{T}}\right)^{\frac{1}{3}}\right) \tag{24}$$

Then the energy and pressure of the ion component of the expanded matter are determined by the relations:

$$\begin{aligned}E_i(\delta,\bar{T}) &= \Lambda\left(\bar{E}_{ex}(\delta) + \bar{E}_{soft}(\delta,\bar{T})\right);\\ P_i(\delta,\bar{T}) &= \Lambda\rho_0\left(\bar{P}_{ex}(\delta) + \Gamma_{soft}(\delta,\bar{T})\delta\bar{E}_{soft}(\delta,\bar{T})\right).\end{aligned} \tag{25}$$

The parameter $Q$ was employed in equations (22)–(24) by D. Young [20] to reduce the influence of the electron heat capacity on the thermodynamic functions of the ion component of liquid metals for defining the parameters of equations of the state according to experimental data. In the proposed approach, we attribute the solid-liquid and liquid-gas PT to the ion component of the matter when thermal excitation is taken into account in the electron liquid (gas) regardless of the thermal excitation of ions (atoms). Therefore, one should use the experimental values of pressure, temperature, and volume jump during melting to determine the values of $Q$ in the solid and liquid states of the expanded metal. For iron: $P = 0.101$ MPa, $T_m = 1811$ K, $\Delta V_m/V_{Lm} = (V_L - V_{Sm})/V_{Lm} = 0.034$, $\rho_{Lm} = 7.020$ g/cm$^3$ [27]. Using it one can find $\rho_{Sm} = 7.26708$ g/cm$^3$. Indexes "S" and "L" denote liquid and solid states, and index "m" denotes melting. Thus, we obtain the following values of $Q$: $Q_{Sm} = 0.548$ for the expanded solid iron; $Q_{Lm} = 0.866$ for its melt. The following expression determines the change in enthalpy in the ion subsystem during the melting of iron:

$$\Delta H_m = \Lambda\left(\bar{P}_m\left(\frac{1}{\delta_{Lm}} - \frac{1}{\delta_{Sm}}\right) + \bar{E}_{soft}(\delta_{Lm},\bar{T}_m) - \bar{E}_{soft}(\delta_{Sm},\bar{T}_m)\right) \tag{26}$$

for $\bar{P}_m = P_m/(\Lambda\rho_0) = 1.731\cdot10^{-6}$, $\delta_{Sm} = {}_{Sm}/\rho_0 = 0.923$, $\rho_{Lm} = \delta_{Lm}/\rho_0 = 0.0891$, $\bar{T}_m = k_BT_m/\Lambda_a = 3.638\cdot10^{-2}$ one can obtain $\Delta H_m = 9.564$ kJ/mol when experimental value is $\Delta H_m = 13.806$ kJ/mol [27]. The conditions of equality to zero at the critical point of the first and second derivatives $\partial P_i(V,T)/\partial V$ and $\partial^2 P_i(V,T)/\partial V^2$ allow one to determine

the values of the critical density. And then, it is possible to find the critical pressure of iron according to the equation of state. Their dimensionless values are the following: $\delta_{cr} = 0.263$, $\overline{T}_{cr} = 0.218$, $\overline{P}_{cr} = 2.152 \cdot 10^{-2}$. Accordingly, the dimensional values of the critical parameters are the following: $\rho_{cr}$ = 2.070 g/cm$^3$; $T_{cr}$ = 10867 K; $P_{cr}$ = 1.256 GP. Index "cr" denotes a critical point.

The critical compressibility factor for pair potential 9-3 is $Z_{cr} = 0.375$, i.e., corresponds to [19].

One can see in Figure 3 the behavior of the Grüneisen parameter of iron for expanded (a) and compressed (b) matter. This parameter for expanded iron at low temperatures and $\delta = 1$ is equal to 3, see the uppermost curve in Figure 1a. This case corresponds to the temperature $T = 1.571 \cdot 10^{-3}$ K. The sharp jump of $\Gamma_{soft}$ is associated with a volume (density) change during melting. It is demonstrated in Figure 3b, the Grüneisen parameter of the compressed metal tends to its limit value of 2/3. This jump decreases as the temperature rises. It is worth noting that at the critical point ($\delta_{cr} = 0.263$), this parameter tends to the value of $\Gamma_{soft} \sim 1$. So it becomes practically independent of temperature (the curves for $T/T_{cr} = 1.04$ and $T/T_{cr} = 1.04$ merge each other). This fact speaks about the non-ideal state of the expanded iron in the area near the critical point. Reduced thermodynamic stability in the region of the critical point of the liquid-gas PT was observed [9,32–34]. Similar intense fluctuations affecting the dynamics of physical processes at supercritical pressures are observed in it (see [3] and refs therein).

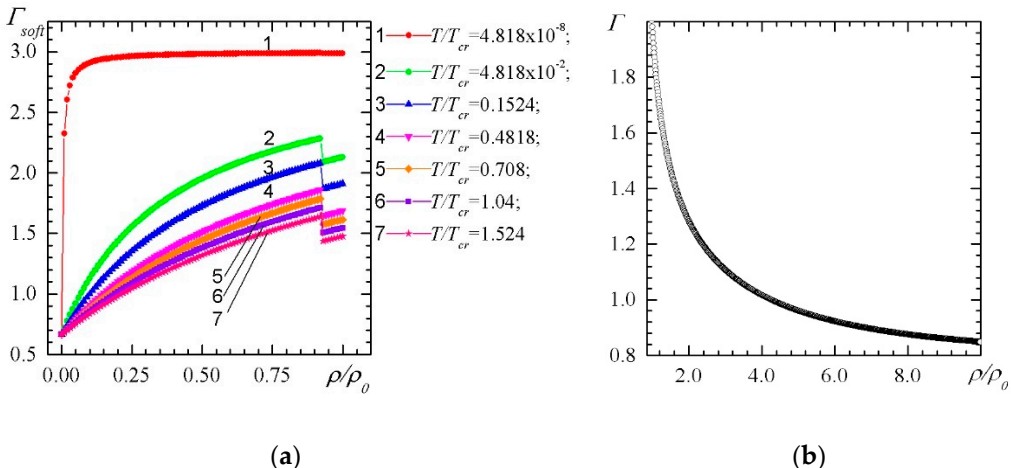

(a)                             (b)

**Figure 3.** Dependence of the Grüneisen parameter on density and temperature for expanded (**a**) and compressed (**b**) iron.

*2.4. Thermodynamic Functions of the Electronic Component of a Metal*

One can express the free energy of an electron gas with a variable number of particles equal to the charge of an atomic cell $z_i$ in the approximation of an average ion as follows:

$$F_e(V_a, T) = z_i \mu_{eff}(V_a, T) + \int_0^Z E_g(z)dz - \frac{(2m)^{3/2} V_a}{3\pi^2 \hbar^3} \int_0^\infty \varepsilon^{3/2} f_0 d\varepsilon \qquad (27)$$

Here $\mu_{eff}$ is the effective chemical potential of the electron gas given by Fermi-Dirac statistics having the following distribution function:

$$f_0 = \left( \exp\left( \frac{\varepsilon - \left( \mu_{eff} - E_g \right)}{T} \right) + 1 \right)^{-1} = \left( \exp\left( \frac{\varepsilon - \mu_1}{T} \right) + 1 \right)^{-1} \qquad (28)$$

here $V_a = 1/n_i$, $m$, $\varepsilon$, $T$ are the volume per one atomic cell (ion), an electron mass, energy, and temperature, respectively.

The energy value $E_g(V_a)$ is the width of the energy gap, which we proposed in our earlier paper [35] for the phenomenological description of the metal-insulator transition at $T = 0$ [4] within the framework of the metal plasma model [36].

Condition for normalizing the Fermi-Dirac distribution function:

$$Z_i(V_a, T) = \frac{2^{1/2}(mT)^{3/2}V_a}{\pi^2\hbar^3}F_{1/2}\left(\frac{\mu_1}{T}\right) \tag{29}$$

$$F_\nu(x) = \int_0^\infty \frac{\xi^\nu d\xi}{\exp(\xi - x) + 1}\ (\nu > -1) \tag{30}$$

defines an implicit relationship between the number of particles and the chemical potential $\mu_1$. The equation (29) can be used to calculate the average ionization composition of a substance in an equilibrium state. It can be done in a wide range of temperatures and volumes if chemical potential dependence on atom cell volume is known [35].

Since one can strictly speak of a metal-insulator PT only at $T = 0$, the energy $E_g$ depends only on the volume of the atomic cell. In the long-wavelength approximation, the electrostatic field does not penetrate the metal. The latter means that the permittivity of the metal formally is infinite. Defining $E_g$ similarly to [35], it can be written as follows:

$$E_g\left(\frac{V_{a0}}{V_a}\right) = E_g(\delta) = \begin{cases} \frac{I_1}{\varepsilon_r(\delta)} = \frac{I_1}{1+3\delta(\delta_* - \delta)^{-1}}, \ \delta \leq \delta_*; \\ 0, \ \delta \geq \delta_*, \end{cases} \tag{31}$$

here $\delta_* = V_*/V_0 = V_{a*}/V_{a0}$ means the relative density of the substance metallization, $I_1$ is the first ionization potential. In Section 7, Chapter III of the classic monograph [7], an approximate calculation of multiple gas ionization is considered in detail. The use of the $I_1/2$ value instead of $I_1$ for the evaluation of reaction constants of a neutral atom lets one obtain the best agreement with the Saha equations calculations in the area of the first ionization potential. In our case, it is also possible to expect that one can obtain the best agreement at $T = 0$ if the chemical potential tends to asymptote $\mu_1 \approx -E_g(\delta)/2$ in the region of $\delta \leq \delta_*$.

The $\delta_*$, $Z_i$ values, and chemical potential for the expanded and compressed states of iron near $\delta = 1$ in the framework of the DFT method using LmtART-7 software package were calculated [21,22]. Using the chemical potential data tables, as well as exchange quantum corrections to them, calculated using the quantum statistical model of the atom (QSM) [26], the chemical potential of iron was recalculated for the region far from $\delta = 1$. After that, the average charge of iron ions in the region of high compressions was calculated for $\delta = 10$–15. In logarithmic coordinates, the data matching was carried out so that the values of the average ion charge (i.e., the number of electrons in an atomic cell) in the intermediate region fit smoothly with the DFT model and QSM calculations. We used the dependence of $\mu_1(V_a, T = 0)$ obtained in this way as an additional correction to $\mu_{QSM}$ for iron. We modified the values of the chemical potential obtained using QSM according to the following:

$$\mu_{QSLMT}(V_a, T) = \mu_{QSM}(V_a, T) - \mu_{QSM}(V_a, T = 0) + \mu_1(V_a, T = 0) \tag{32}$$

And one can calculate the thermal energy

$$\varepsilon_{Te}(V_a, T) = \begin{cases} Z_i(V_a, T)T\frac{F_{3/2}(x)}{F_{1/2}(x)} - \frac{3}{5}Z_{ic}(V_a)\mu_c(V_a), \ V_a \leq V_{a*}, \\ \qquad Z_{ic} = Z_i(V_a, 0), \mu_c = \mu_{QSLTM}(V_a, 0); \\ Z_i(V_a, T)T\frac{F_{3/2}(x)}{F_{1/2}(x)}, \ V_a > V_{a*} \end{cases} \tag{33}$$

and thermal pressure as follows:

$$P_{Te}(V_a, T) = \frac{2}{3} \frac{\varepsilon_{Te}(V_a, T)}{V_a} = \frac{2}{3} \rho \frac{\varepsilon_{Te}}{M_a} = \Lambda \rho_0 \frac{2}{3} \delta \tilde{\varepsilon}_{Te}, \tag{34}$$

It was possible to define the number of electrons (average ion charge) in an atomic cell $Z_i(V_a, T)$ according to (29) in a wide range of specific volumes and temperatures. Figure 4 shows the calculation results. Solid and dashed lines show the boundaries of the solid-liquid and metal-insulator PT, respectively.

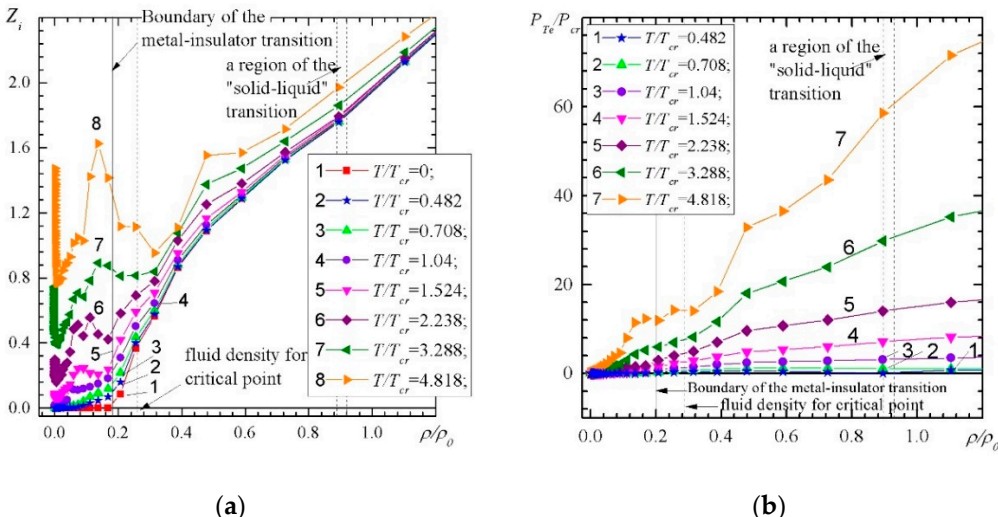

(a)  (b)

**Figure 4.** Isotherms of average ion charge (**a**) and thermal electron pressure (**b**).

The behavior of the average charge of the ion in comparison with the experimental results of [3] is discussed below.

The distinguishing feature of the approach we used is that a change in the metallic conductivity of a matter to a non-metallic one due to the chaos growth is considered the metal-dielectric phase transition [37]. It occurs in the supercritical region of expanded iron in the range of relative volume $V/V_0 = 0.3$–$0.4$ (i.e., densities $\delta = 0.25$–$0.333$).

As shown in Figure 4a, this range includes the critical iron density $\delta = 0.263$ obtained in Section 2.3 above (dashed line). Also, the experimental pressure and temperature values exceed the parameters of the critical point. The lower curve in Figure 4a corresponds to $T = 0$. DFT calculations using the LmtART-7 software package [21,22] gave the density $\rho_{MIT} = 1.432$ g/cm$^3$ ($\delta_{MIT} = 0.182$). It corresponds to the metal-insulator PT at $T = 0$ when the overlap of wave functions disappears, i.e., collectivized electrons vanish. At a density of $\delta = 1$ and $T = 0$, the number of quasi-free ("valence") electrons is equal to $Z_{ic} = 1.95$. At the same time, the total number of iron valence electrons taken into account in the DFT method is 8. In the region of densities $\delta_{MIT} < 0.182$, electrons obey classical statistics, and at high densities, Fermi statistics. The peculiarities of $Z_i$ observed in the left part of the plot for lower $\delta$ are caused by the competition between the mechanisms associated with pressure and temperature. The point is that in the region of $0.01 < \delta < 0.136$, a temperature increase leads to a $Z_i$ increase, on the one hand. And on the other hand, the energy gap $E_g(\delta)$ increases due to a density decrease and leads to a slowdown in the ionization rate (see (31)). The increase of $Z_i$ for supercritical temperatures and very low densities occurs due to these densities corresponding to an ideal gas. After all, the minimum density is $\delta_{min} = 2.0674 \times 10^{-6}$ ($\rho_{min} = 1.627 \times 10^{-5}$, g/cm$^3$). In comparison to $Z_i$ curves, the curves in Figure 4b show a much more monotonic behavior of the $P_{Te}/P_{cr}$ isotherms as a function of $\rho/\rho_0$ in the supercritical region. To define the contribution of the thermal electron pressure to the total pressure, we should make some additional remarks about the metal-dielectric PT and describe methods of the electrical conductivity in the region of the compressed and

expanded states of a metal. We can obtain the following expression of conductivity using the results of the previous works [8,35]:

$$\sigma = \frac{\sqrt{2}e^2 Z_i n_i r_s \frac{\partial}{\partial x}\int_0^\infty \xi l_{eff}(\xi) f_0(\xi - x)d\xi}{3\sqrt{mT}\, F_{1/2}(x)} \tag{35}$$

here $r_s = (3/4\pi n_i)^{1/3}$ and $l_{eff}(\xi) = (1 + A^2\xi^4/(Z_i^2 L_1 - 2Z_i(Z - Z_i)L_2 + (Z - Z_i)^2 L_3)^2)^{1/2}$ is mean dimensionless effective electron free path, in which $Z$ is a nuclear charge, $A = GT^2/(\pi n_i e^4 r_s)$; $G = \rho\partial P_i/\partial\rho(n_i T)^{-1}$ is the structural factor of the metal in the long-wave approximation; $L_1$, $L_2$, and $L_3$ are Coulomb logarithm analogs defined as follows:

$$L_1(4k^2/k_D^2) = \tfrac{1}{2}\left(\ln\left(1 + \frac{k^2}{k_D^2}\right) - \frac{4k^2 k_D^{-2}}{1 + 4k^2 k_D^{-2}}\right),$$
$$L_2 = \tfrac{1}{2}\ln\left(1 + 4k^2 r_{cd}^2\right)\left(\frac{k_D^2 r_{cd}^2}{k_D^2 r_{cd}^2 - 1}\ln\frac{1 + 4k^2 k_D^{-2}}{1 + 4k^2 r_{cD}^2} - \frac{1}{k_D^2 r_{cd}^2 - 1}\right), L_3 = L_1\left(4k^2 r_{cd}^2\right); \tag{36}$$

here $r_{cd} = r_c/(1 + k_D r_c)$; $r_c$—shielding radius of an ion (atom) having $Z_i = 0$; $k_d = \min\{k_s^2; k_{ei}^2\}$; $k_{ei}^2 = k_{De}^2 + (k_{Di}^2 + k_e^4)k_{Di}^{-2}$; $k_{Di}^2 = 4\pi e^2 Z_i^2 n_i/T$; $k_{De}^2 = 2\pi e^2 Z_i n_i T^{-1}F_{-1/2}(x)/F_{1/2}(x)$. The structural factor $G$ in the mean free path considers the electrons scatter in metal, similar to that of X-ray waves on density fluctuations. This approach allowed J. I. Frenkel to explain the dependence of the metal conductivity on temperature: $\sigma \sim T^{-1}$ [38]. In a metal at $\delta \geq 1$, the conductivity changes continuously [8,35]. So the metallic dependence of conductivity on temperature persists up to $T \sim T_{cr}$. After removing the degeneracy of the electronic component, the conductivity dependence changes to plasma one: $\sigma \sim T^{3/2}$. In the intermediate temperature range at a constant density, the conductivity is minimal and practically does not depend on temperature. In the region of expanded matter much higher than the critical temperature, one can expect the conductivity changes smoothly from a metallic type to a plasma one. Below the critical density in the region of the liquid-gas PT for the ionic component of the expanded matter, the dependence is $P_i/P_{cr} = f(T/T_{cr})$. Therefore, the structure factor in the two-phase region and the critical point is $G = 0$ and $l_{ef}f = 1$.

Then relation (35) will be as follows:

$$\sigma = \frac{\sqrt{2}e^2 Z_i(n_i, T)n_i r_s(n_i)}{3\sqrt{mT}}\frac{F_1'(x)}{F_{1/2}(x)} \tag{37}$$

Also, the relation (37) can be used for estimates in the near critical point and in the overcritical region, the so-called supercritical fluid [33,34], which is characterized by low stability of matter and high fluctuations of density. We plan to publish a detailed study of physical processes in this area in a separate paper.

## 3. Discussion

Let us regard the behavior of the thermodynamic functions of the expanded state of iron, considering the thermal excitation of the electronic component.

Figure 5 shows the binodal along with the dependence $P_b/P_{cr} = f(T/T_{cr})$ (a), as well as the pressure of the ion component of the expanded metal (b). One can see that the pressure of the ion component appears to be positive only at $T/T_{cr} > 0.78$, which corresponds to the value $\delta_L = \rho_L/\rho_0 = 0.54$.

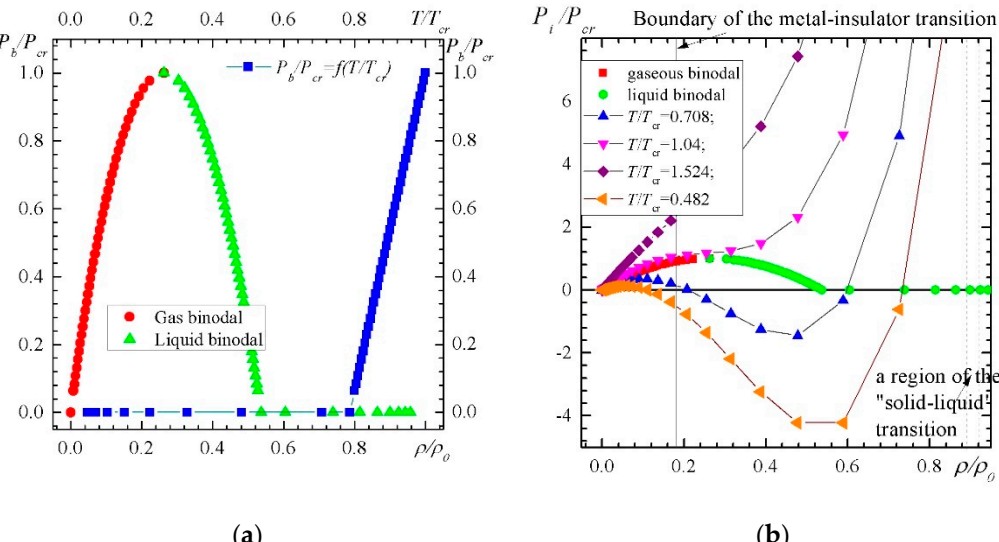

**Figure 5.** Binodal and $P_b/P_{cr} = f(T/T_{cr})$ (**a**), pressure of the ion component $P_i/P_{cr} = f(T/T_{cr})$ (**b**).

The results of calculations of the total pressure of the expanded metal in dependence on the relative matter density are shown in Figure 6. As one can see, the thermal pressure of electrons $P_{Te}/P_{cr}$ shifts the intersection of $T/T_{cr} = 0.707$ isotherm by the binodal branch of the liquid state matter to the positive region pressures of $P_\Sigma/P_{cr}$.

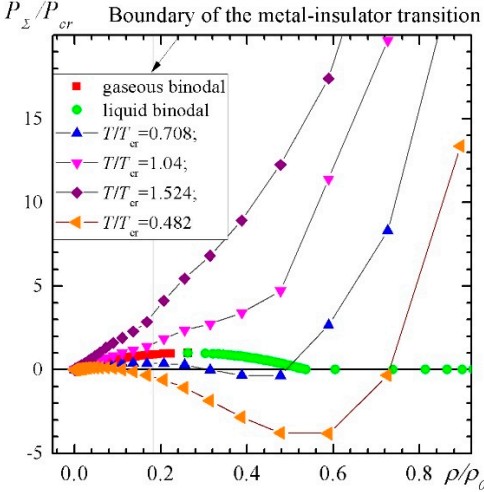

**Figure 6.** The total pressure of the expanded metal vs relative density.

At the same time, the total pressure for the $T/T_{cr} = 0.482$ isotherm changed not significantly because the thermal pressure of the electronic component is significantly less than that for ions, according to the data presented in Figure 4b.

Total and ion pressures at $\delta = \rho/\rho_0 \leq 0.18$ practically coincide for $T/T_{cr} = 1.04$. At higher densities, the total pressure isotherm is higher, but the qualitative character of its behavior in the near-critical region is close to that of the ionic pressure. Since we are dealing with a non-ideal low-temperature plasma [5,6], further improvement of the new critical parameters to consider the contribution of thermal electrons to the total pressure is not required. Indeed, according to our calculations, the average charge of the ion component of iron at the critical point is approximately 0.5, i.e., we deal with a mixture of neutral atoms and singly charged ions.

## 4. Conclusions

The discussion above allows us to conclude that the proposed expressions are usable for modeling the high-energy effects on metal. In particular, they can be used to describe the influence of pulsed laser radiation on electrodes placed in an electric field for modeling the plasma evolution of laser triggered switches. The value of the critical volume $V/V_0 = 3.802$ obtained according to the proposed model is well consistent with the experimentally established range of the change in the metallic type of conduction to plasma one [3].

**Author Contributions:** Conceptualization, N.B.V.; Investigation, A.I.L. All authors have read and agreed to the published version of the manuscript.

**Funding:** This research was funded by Russian Science Foundation and Government of Sverdlovsk Region, project No. 22-29-20058.

**Data Availability Statement:** The data that support the findings of this study are available upon reasonable request from the authors.

**Acknowledgments:** We are grateful to S.Y. Savrasov for the opportunity to use the LmtART-7 software package for the calculation of the electronic spectra of iron in the framework of the density functional theory.

**Conflicts of Interest:** The authors declare no conflict of interest.

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
