# Peer review of "Thermodynamic Functions of a Metal Exposed to High Energy Densities in Compressed and Expanded States"

_condensedmatter, doi:10.3390/condmat7040061_

Round 1

Reviewer 1 Report

From the scientific point of view your paper is interesting and useful. A question: do you see applications of your results in DAC experiments,or in astrophysics? As it is,your paper can not be published.You must improve the language.The text is full of small errors. I propose that you carefully correct the language and then submit again

Author Response

Dear Reviewer!

  1. We have checked our English
  2. We have extended introduction
  3. As for the question about the Bridgman diamond anvils technique in a similar one, of course, we hope our results can be applied to such experimental data along with data obtained from experiments concerning laser pulsed radiation interaction with metals.

Reviewer 2 Report

Report on manuscript No. 1971330

This is a good work leading to a description of a wide range EOS for metals. It covers highly compressed states, expanded states, liquid-vapor transition and metal-insulator transition. The following points are noteworthy: 1) correction to Born-Mayer potential for strong compression, 2) incorporating metal-insulator transition in the expanded states. 3) Use of electron-chemical potential from DFT calculations to determine the free-election density,  and 4) incorporating electron-energy gap due to the metal-insulator transition are good efforts.

The method described would be of  use to researchers engaged in applications of wide-range EOS of metals.

I recommend publication of the paper in Condensed Matter.

The following are a partial list of corrections the authors may make in the manuscript.

1. line 119:  defining may be changed to definition

2. line 155: the statement regarding eq. 2.5, 2.6 2.8 and 2.9 is not clear.

3. line: 242: D.Yang to be changed to D.Young

4. line: 273: figure 1 may be changed to figure 3.

5: line 295: change is to are

6. line: 307  remove in

7. line 352:  change a region to region in figure captions. fig-4 & fig-5

8. line 375: change description to describe 

9. Is it necessary to give 15 decimal places for the numbers ? Not sure whether these numbers are so accurate or not.

10. Please try to check for other spellings.

Author Response

Dear Reviewer!

Thank you for review.

Point 1: line 119:  defining may be changed to definition 

Corrected

Point 2. line 155: the statement regarding eq. 2.5, 2.6 2.8 and 2.9 is not clear.

Sentence rewritten

Point 3. line: 242: D.Yang to be changed to D.Young

Corrected

Point 4. line: 273: figure 1 may be changed to figure 3.

Corrected

Point 5. line 295: change is to are

Corrected

Point 6. line: 307  remove in

Removed

Point 7. line 352:  change a region to region in figure captions. fig-4 & fig-5

The range 0.25-0.333 indicates only a supercritical region, whereas figures show the full calculated range.

Point 8. line 375: change description to describe 

Changed

Point 9. Is it necessary to give 15 decimal places for the numbers ? Not sure whether these numbers are so accurate or not.

We reduced shown decimal places

Point 10. Please try to check for other spellings.

We did a deep spell check.
